:ᨆ: PLOS | ONE

# Associations between high blood pressure and DNA methylation

Nabila Kazmi[1,2]*, Hannah R. Elliott[1,2], Kim Burrows[1,2], Therese Tillin[3], Alun D. Hughes[3,4], Nish Chaturvedi[3,4], Tom R. Gaunt[1,2,5ʘ], Caroline L. Relton[1,2,5ʘ]

1 MRC Integrative Epidemiology Unit, University of Bristol, Bristol, United Kingdom, 2 Population Health Sciences, Bristol Medical School, University of Bristol, Bristol, United Kingdom, 3 Department of Population Science & Experimental Medicine, Institute of Cardiovascular Science, University College London, London, United Kingdom, 4 MRC Lifelong Health & Aging Unit at UCL, London, United Kingdom, 5 NIHR Bristol Biomedical Research Centre, Bristol, United Kingdom

ʘ These authors contributed equally to this work.
* nabila.kazmi@bristol.ac.uk

## Abstract

### Background

High blood pressure is a major risk factor for cardiovascular disease and is influenced by both environmental and genetic factors. Epigenetic processes including DNA methylation potentially mediate the relationship between genetic factors, the environment and cardiovascular disease. Despite an increased risk of hypertension and cardiovascular disease in individuals of South Asians compared to Europeans, it is not clear whether associations between blood pressure and DNA methylation differ between these groups.

### Methods

We performed an epigenome-wide association study and differentially methylated region (DMR) analysis to identify DNA methylation sites and regions that were associated with systolic blood pressure, diastolic blood pressure and hypertension. We analyzed samples from 364 European and 348 South Asian men (first generation migrants to the UK) from the Southall And Brent REvisited cohort, measuring DNA methylation from blood using the Illumina Infinium® HumanMethylation450 BeadChip.

### Results

One CpG site was found to be associated with DBP in trans-ancestry analyses (i.e. both ethnic groups combined), while in Europeans alone seven CpG sites were associated with DBP. No associations were identified between DNA methylation and either SBP or hypertension. Comparison of effect sizes between South Asian and European EWAS for DBP, SBP and hypertension revealed little concordance between analyses. DMR analysis identified several regions with known relationships with CVD and its risk factors.

**Data Availability Statement:** Data supporting the results reported in this paper can be found in the Supporting Information files. The authors are unable to make individual level data available due to

concerns regarding compromising individual privacy. However, individual level data are available on request from the SABRE Study at University College London via email at sabre@ucl.ac.uk.

**Funding:** The SABRE study was funded at baseline by the Medical Research Council, Diabetes UK, and the British Heart Foundation. At follow-up the study was funded by the Wellcome Trust and the British Heart Foundation. Methylation analysis in the SABRE cohort was supported by a Wellcome Trust Enhancement grant (082464/Z/07/C). AH and NC received support from the Medical Research Council (Programme Code MC_UU_12019/1) and the National Institute for Health Research University College London Hospitals Biomedical Research Centre. The SABRE study team also acknowledges the support of the National Institute of Health Research Clinical Research Network (NIHR CRN). NK, HRE, TRG and CLR are supported by the Medical Research Council Integrative Epidemiology Unit at the University of Bristol (MC_UU_12013/2 and MC_UU_12013/8). The SABRE study team also acknowledges the support of the National Institute of Health Research Clinical Research Network (NIHR CRN).

**Competing interests:** TRG receives funding from GlaxoSmithKline, Biogen and Sanofi for research unrelated to the work presented here. This does not alter our adherence to PLOS ONE policies on sharing data and materials.

**Abbreviations:** BH, Benjamini and Hochberg; BP, Blood pressure; BMI, Body mass index; CVD, Cardiovascular disease; DBP, Diastolic blood pressure; DMR, Differentially methylated region; EWAS, Epigenome-wide association study; FDR, False discovery rate; HM450, HumanMethylation450; MAF, Minor-allele frequency; NK, Natural killer; PCA, Principal Component Analysis; PCs, Principal components; SABRE, Southall and Brent Revisited; SBP, Systolic blood pressure.

## Conclusion

This study identified differentially methylated sites and regions associated with blood pressure and revealed ethnic differences in these associations. These findings may point to molecular pathways which may explain the elevated cardiovascular disease risk experienced by those of South Asian ancestry when compared to Europeans.

## Introduction

Hypertension results from abnormalities in the control systems that normally regulate blood pressure [1]. It is one of the strongest risk factors for cardiovascular disease (CVD) which is the leading cause of death worldwide [2, 3]. People of South Asian descent have increased risk of CVD compared to Europeans [4, 5]. In South Asians living in the United Kingdom, death rate from stroke is between 20% and 25% greater than the rest of the population [6]. Associations between SBP or DBP and stroke are also stronger in South Asians than Europeans [4].

DNA methylation is an example of an environmentally responsive, mitotically stable, epigenetic mark that is associated with biological processes, including those leading to high blood pressure and stroke [7–9]. Furthermore, candidate gene analyses in cell-line and animal studies have demonstrated a role of DNA methylation in the pathogenesis of hypertension [10–14]. For one of these genes, *HSD11B2*, DNA methylation has also been associated with hypertension in humans [15]. One recent study found associations between systolic and diastolic BP and DNA methylation among participants of European, Hispanic and African American decent [16]. We aimed to identify DNA methylation associated with SBP, DBP and hypertension in peripheral blood of European and South Asian men in data collected using the Infinium HumanMethylation450 (HM450) BeadChip. We analyzed all samples together in a trans-ancestry analysis, then conducted analyses in each ethnic group separately. We hypothesised that BP associated epigenetic marks would differ between South Asian and European groups, highlighting potential mechanisms explaining the disparity in CVD and stroke risk between the two ethnicities.

## Results

### Trans-ancestry EWAS analyses

**SBP.**   In the unadjusted EWAS investigating the association between DNA methylation and SBP we identified four CpG sites with Bonferroni-corrected p-value$< 1.24\times10^{-07}$ (Table A in S1 File). After adjustment for potential confounders these associations were markedly attenuated (Table B in S1 File).

**DBP.**   In the unadjusted EWAS investigating the association between DNA methylation and DBP we identified two CpG sites, with Bonferroni-corrected p-value (Table C in S1 File). After adjustment for potential confounders, the associations identified in the unadjusted EWAS were markedly attenuated. However, one additional CpG site (cg07598370 near *OR5AP2)* was identified in the adjusted EWAS, below the Bonferroni-corrected threshold (Fig 1 and Table D in S1 File). DBP was associated with lower DNA methylation at this CpG site with a 0.1% decrease in DNA methylation per 1 mmHg increase in DBP. The mean and standard deviation (SD) of DNA methylation at this CpG site in the European group was 0.84 (0.07) and in the South Asian group was 0.83 (0.05).

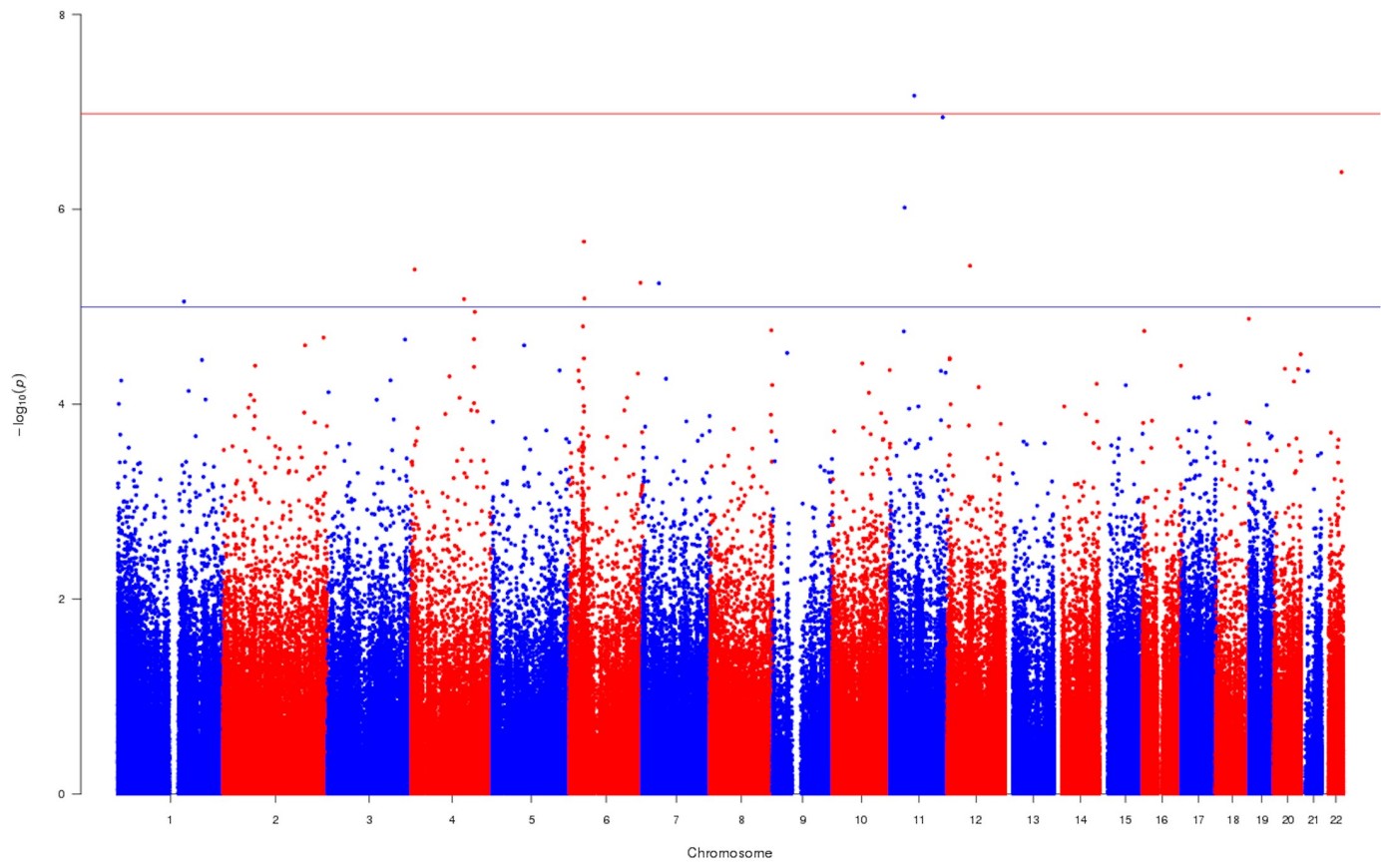

**Fig 1. Manhattan plot indicating the associations between DBP and DNA methylation of men of European and South Asian ancestry.** The plot demonstrates the associations between DBP and DNA methylation of European and South Asian men collectively. The model was adjusted for confounders, estimated cell counts and principal components. The uncorrected log10(p-values) are plotted on the y-axis. The blue line is drawn to separate the CpG sites that surpassed p-value<1×10$^{-05}$, a threshold for a suggestive association and the red line to separate the CpG sites that surpassed the Bonferroni-corrected threshold (p-value<1.24×10$^{-07}$). The CpG sites that surpassed the Bonferroni-corrected threshold were considered to be associated with the trait.

**Hypertension.** In the unadjusted EWAS investigating the association between DNA methylation and hypertension, we identified two CpG sites, falling below the Bonferroni-corrected p-value threshold (Table E in S1 File). After adjustment for potential confounders, the associations identified in the unadjusted EWAS were markedly attenuated and no longer associated (Table F in S1 File).

## European EWAS

**SBP.** In the EWAS investigating the association between SBP and DNA methylation, no CpG sites were associated with SBP after correction for multiple testing (G and H Tables in S1 File).

**DBP.** In the unadjusted EWAS investigating the association between DNA methylation and DBP, we identified three CpG sites with Bonferroni-corrected p-value<1.24×10$^{-07}$ (Table I in S1 File). In the fully adjusted EWAS, two of the three CpG sites: cg16241714 (in the gene body of *CEBPD*), cg00006122 (near the gene *C12orf44*), were identified as falling below the Bonferroni-corrected threshold (Table J in S1 File). Their direction of association was consistent with the unadjusted EWAS. In addition to these two CpG sites, cg04751533 near

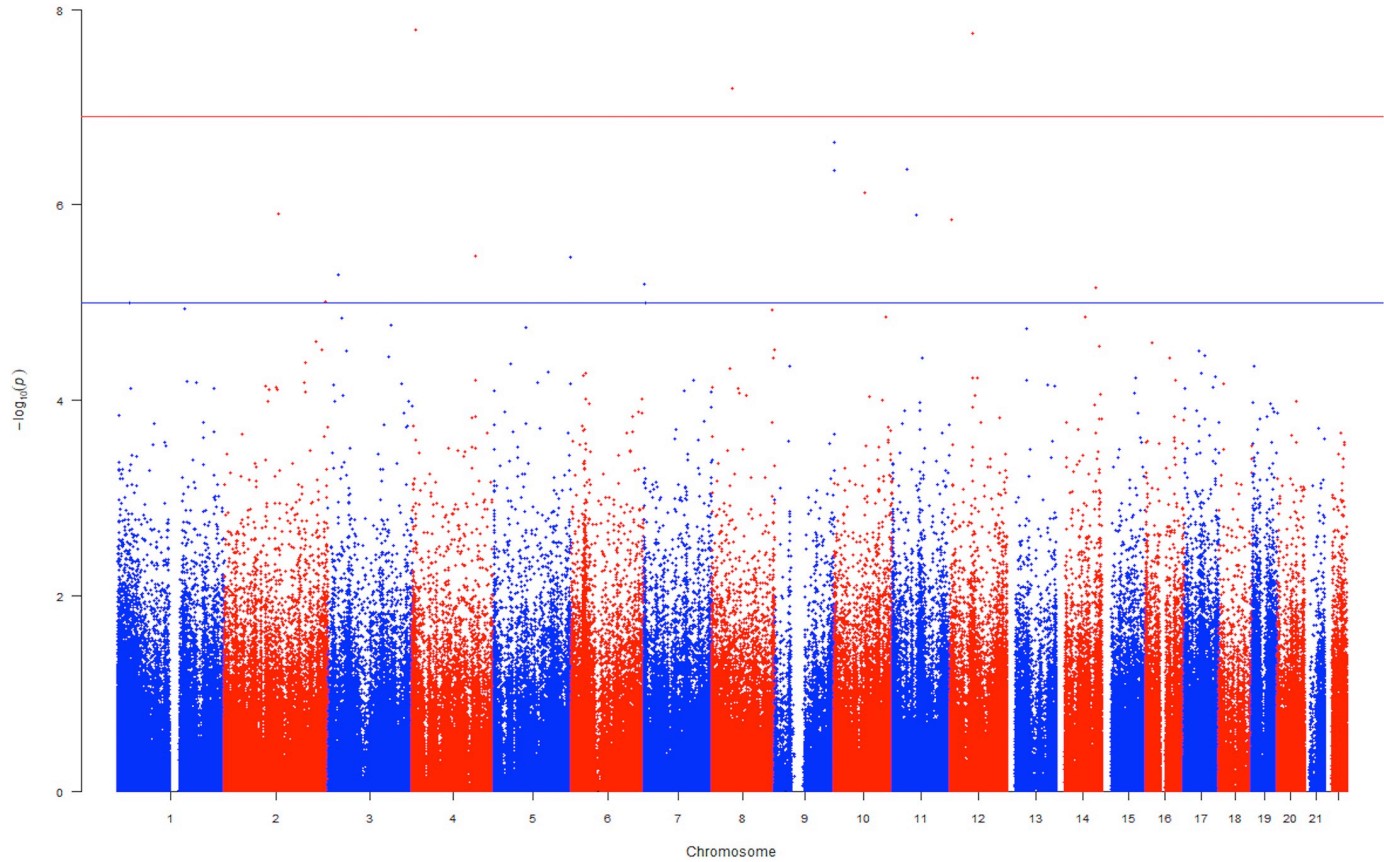

**Fig 2. A Manhattan plot indicating the associations between DBP and DNA methylation in European men.** The plot demonstrates the associations between DBP and DNA methylation in European men. The model was adjusted for confounders, estimated cell counts and principal components. The uncorrected–log10 (p-values) are plotted on the y-axis. The blue line is drawn to separate the CpG sites that surpassed p-value<1×10$^{-05}$, the threshold suggestive of an association and the red line to separate the CpG sites that surpassed the Bonferroni threshold (p-value<1.24×10$^{-07}$). The CpG sites that surpassed the Bonferroni threshold were considered to be associated with the trait.

*AFAP1* was found to be associated with DBP a 0.16% decrease in DNA methylation per 1 mmHg increase in DBP observed at this CpG site (Fig 2 and Table J in S1 File).

**Hypertension.** In theEWAS investigating the association between hypertension and DNA methylation, no CpG sites were associated with hypertension after correction for multiple testing (K and L Tables in S1 File).

## South Asian EWAS

**SBP.** In the unadjusted EWAS investigating the association between DNA methylation and SBP, we identified one CpG site (cg07963349 in the gene body of *GALR2)*, which fell below the Bonferroni-corrected p-value<1.24×10$^{-07}$ (Table M in S1 File). In the fully adjusted EWAS, this CpG site was no longer associated after correction for multiple testing but the direction of association was consistent between models (Table N in S1 File).

**DBP.** In the EWAS investigating the association between DBP and DNA methylation, no CpG site was associated following correction for multiple testing (O and P Tables in S1 File).

**Hypertension.** In both unadjusted and fully adjusted EWAS investigating the association between hypertension and DNA methylation, no CpG site was associated following correction for multiple testing (Q and R Tables in S1 File).

**Table 1. Levels of genomic inflation (Lambda) for each EWAS comparison after correcting for confounders including cell counts.**

|  | DBP | SBP | Hypertension |
|---|---|---|---|
| **Trans-ancestry** | 1.07 | 0.94 | 1.12 |
| **European** | 1.08 | 1.00 | 0.92 |
| **South Asian** | 1.00 | 0.94 | 0.99 |

The genomic inflation factor, lambda, for all fully adjusted models is provided in Table 1.

## Comparison of effect sizes between trans-ancestry, European and South Asian EWAS

The magnitude of associations between European and South Asian populations taken from the fully adjusted EWAS of SBP, DBP and hypertension were compared to each other using a linear fit and a random intercept and slope multilevel model in a pair-wise comparison. There was little consistency between the EWAS of Europeans and South Asians (goodness-of-fit $R^2 = 0$ for SBP, DBP and hypertension (Fig 3)).

## Differentially methylated region analysis

DMR analyses were carried out to identify regions of DNA methylation that were associated with SBP, DBP and hypertension. As in the EWAS analyses, DMR analysis was conducted for the full trans-ancestry group, and additionally for European and South Asian sub-groups. Trans-ancestry DMR analysis identified 395 regions of methylation variation (mapped to 326 annotated genes) for SBP, 237 regions (mapped to 157 annotated genes) for DBP and 0 DMRs for hypertension using a FDR-corrected p-value<0.05 (S-XTables in S1 File). Twelve DMRs annotated to genes were common between trans-ancestry SBP and trans-ancestry DBP at FDR p-value<0.05 (Table 2).

In Europeans, 348 DMRS for SBP (mapped to 291 annotated genes), 95 for DBP (mapped to 74 annotated genes) and 0 for hypertension were identified using a FDR-corrected p-value<0.05.

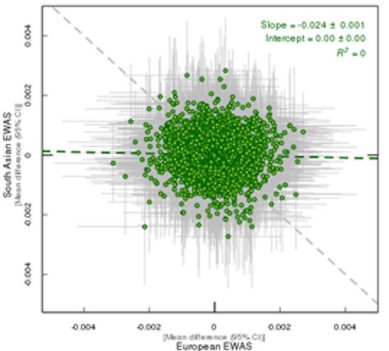
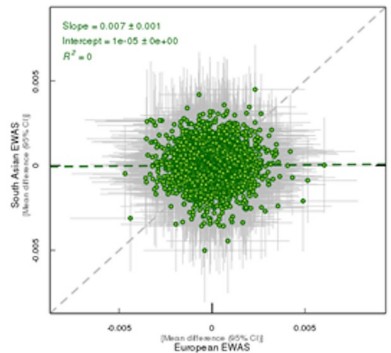
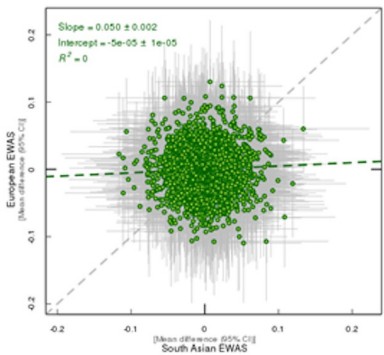

**Fig 3. Consistency between fully adjusted EWAS of SBP, DBP and hypertension between Europeans and South Asian.** DNA methylation associations are shown for 402331 common CpG sites across three EWAS analyses. Each green dot represents a CpG site and the positions of the dots are determined by the effect size in each analysis. The grey lines on each dot denote the confidence intervals (CI) for the estimates. A linear fit of the overall correspondence summarises correlation between compared associations (green dashed line). Grey dashed line shows the line of equality in effect sizes between pairs of analyses. $R^2$ = goodness of linear fit and as such is a measure of the consistency between two EWAS.

**Table 2. Overlap of DMRs between fully adjusted DMR analysis of SBP and DBP for trans-ancestry, Europeans and South Asians respectively.**

| Overlap for trans-ancestry | Overlap for Europeans | Overlap for South Asians |
|---|---|---|
| BBS1* | AR | ACCN3 |
| BCORL1* | ARHGEF3 | ALDH16A1 |
| CACNA1A** | BBS1 | ARID1B |
| HLX* | BCORL1 | ARPP-21 |
| MYT1L | FHL1 | ATP5G3 |
| PDZD2 | HLX | BBS2 |
| PRCC | PRDM16 | C17orf80, FAM104A |
| PRRT1 | SLMO1 | C17orf96 |
| SLMO1* | | C5orf13 |
| TFAP2D | | CACNA1A |
| ZNF77** | | CALHM1 |
| ZNF783 | | CHD4 |
| | | CYFIP1 |
| | | DERA |
| | | ELL2 |
| | | ERF |
| | | IFFO1 |
| | | INPP5A |
| | | LRCH2 |
| | | MSRB3 |
| | | MYH9 |
| | | ORC5L |
| | | P4HA3 |
| | | PID1 |
| | | PPIL6 |
| | | PRKAG2 |
| | | PTPRN2 |
| | | PURG,WRN |
| | | SNORD113-7 |
| | | TRPS1 |
| | | WDR27 |
| | | WIPF1 |
| | | ZIC1 |
| | | ZNF57 |
| | | ZNF77 |

* when a gene is common between trans-ancestry and European analyses,

** when a gene is common between trans-ancestry and South Asian analyses.

Nine DMRs annotated to genes were common between European SBP and European DBP at FDR p-value<0.05 (Table 2). In South Asians, 96 DMRs for SBP (mapped to 66 annotated genes), 186 DBP (mapped to 135 annotated genes) and 0 hypertension DMRs were identified using a FDR-corrected p-value<0.05. Thirty-five DMRs annotated to genes were common between South Asian SBP and South Asian DBP at FDR p-value<0.05 (Table 2).

We created Venn diagrams to identify overlap between DMRs (mapped to genes) of SBP and DBP analyses conducted for South Asian and European groups (Fig 4). For SBP, there

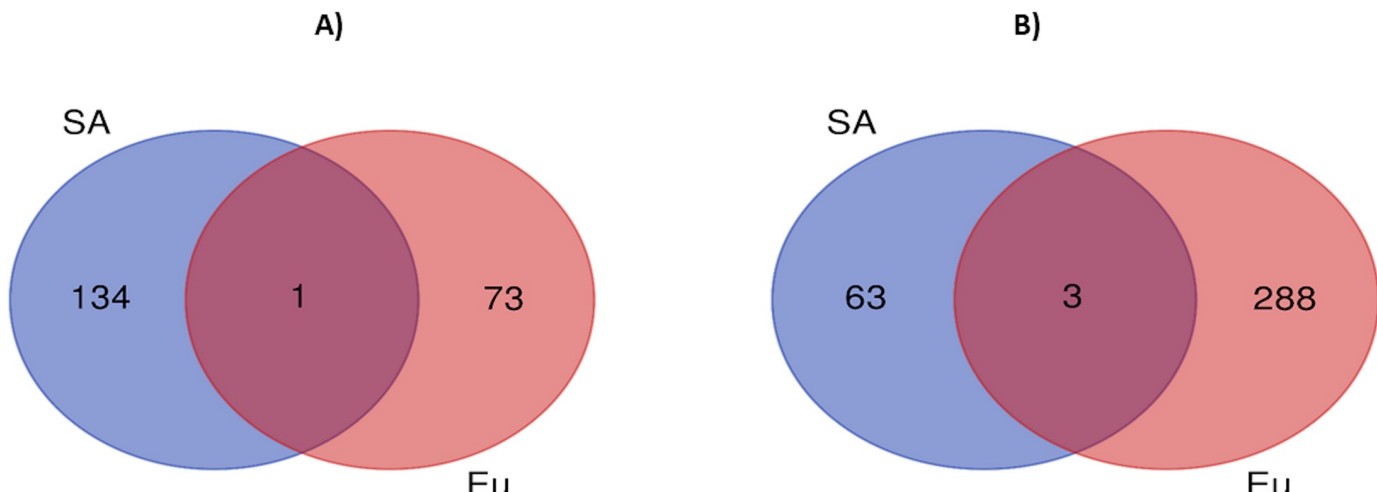

**Fig 4. Venn diagrams to identify overlap between DMRs (mapped to genes) of DBP (A) and SBP (B) analyses conducted for South Asian and European groups.** SA denotes South Asian and Eu denotes European ancestry.

were three genes in common (*WRN*, *PTPRN2* and *CACNA1A*). For DBP, there was one DMR in common between South Asian and European groups (PRDM16).

We performed pathway analyses for the genes mapped for DMRs of SBP and DBP analyses conducted for South Asian and European groups. For SBP analyses conducted for Europeans and DBP analyses conducted for South Asians, we identified pathways enriched with FDR q-value<0.05. The identified pathways for SBP analyses conducted for Europeans were NOTCH2 intracellular domain regulates transcription and NOTCH4 intracellular domain regulates transcription. The identified pathway for DBP analyses conducted for South Asians was insulin-like growth factor-2 mRNA binding proteins (IGF2BPs/IMPs/VICKZs) bind RNA.

**Known genetic variants in DMRs.** Two of the twelve DMRs common in trans-ancestry SBP and DBP analysis, *TFAP2D* and *HLX*, contain SNPs previously associated with blood pressure [17, 18]. The genetic variant in the *PDZD2* DMR was previously associated with myocardial infarction [18]. *HLX* was also identified in DMR analysis of SBP and DBP in Europeans (Table 2). *PRDM16* was found to be common to SBP and DBP in Europeans, its genetic variants were previously associated with dilated cardiomyopathy [19]. *PRKAG2* was found to be common to SBP and DBP in South Asians, and has been previously associated with hypertrophic cardiomyopathy [20] and chronic kidney disease [21]. Other identified genes previously reported in GWAS to be associated with CVD-related traits were *ELL2* (GWAS of BP [17], insulin [22] and glucose [22]), *TRPS1* (GWAS of BP [17]), *PID1* (GWAS of stroke [23], lung function [24] and chronic obstructive pulmonary disease [25]) and *WIPF1* (GWAS of resting heart rate [26]).

## Comparison with previous EWAS

We looked up 126 associations reported earlier [16] that also appeared in the trans-ancestry EWAS of SBP and DBP in our study (Table Y in S1 File). The evidence of associations for these sites was generally weaker in our study. The effect sizes were heterogeneous for the majority of associations but the magnitude of difference between two studies was small. cg19693031 (near *TXNIP*) and cg18120259 (in gene body *LOC100132354*), found within the top hundred CpG sites of the trans-ancestry EWAS of SBP in our analyses, were among the 126 previously reported associations [16]. The direction of effect was consistent with the previous analysis and the magnitude of association was slightly larger in our study. The

above reported cg18120259 was also the top CpG site in our trans-ancestry EWAS of hypertension, the direction of effect was consistent but the magnitude of association was stronger in our study. These CpG sites were also replicated previously [16].

## Discussion

We investigated the association between SBP, DBP and hypertension and DNA methylation measured in peripheral blood of European and South Asian men combined and then across individual ethnicities using the HM450 BeadChip array. In the trans-ancestry fully adjusted EWAS, we found DBP was associated with methylation at one CpG site (cg07598370 near *OR5AP2*) below the Bonferroni-corrected threshold. A genetic variant near the olfactory receptor, family 5, subfamily AP, member 2 (*OR5AP2*) has been previously reported to be associated with hematological phenotypes [27], in addition olfactory receptors are known to regulate blood pressure via their renal expression [28]. Evidence has shown that renal sensory receptors play an important role in blood pressure regulation and olfactory receptors belonging to this group of sensory receptors[29]. SBP and hypertension were not associated with DNA methylation after adjustment for confounders, estimated cell counts and PCs. EWAS were also conducted in European and South Asian groups separately. In Europeans fully adjusted EWAS, three CpG sites were associated with DBP with a Bonferroni-corrected p-value below the threshold imposed. In South Asians fully adjusted EWAS, SBP, DBP and hypertension were not associated with DNA methylation after multiple testing corrections. Several of the initial associations observed in the unadjusted model of DBP were noted to be documented loci responsive to tobacco smoking CpG sites, for example, (cg05575921 [30], cg12803068 [31], cg03636183 [32], cg22132788 [33] and cg09935388 [34]), hence their attenuation on adjustment for smoking was predictable. This highlights the capacity of DNA methylation to index exposure to relevant risk factors.

We found three CpG sites associated with DBP in Europeans. One CpG site (cg04751533), in the gene body of *AFAP* (actin filament-associated protein), acts as an actin-binding and crosslinking protein and is enriched in SRC and phorbol ester induced podosomes [35]. Podosomes are specialized plasma-membrane actin-based microdomains and have been suggested to play a role in arterial vessel remodeling [36]. *C12orf44* (cg00006122) encodes *a*utophagy-related protein 101 that is also known as *ATG101*. *ATG101* is an autophagy related protein, with relevance to blood pressure because autophagy plays a key role in pulmonary vascular remodelling via regulation of apoptosis and hyperproliferation of pulmonary arterial endothelial cells [37]. *ATG101* is an essential gene for the initiation of autophagy and may be involved in endothelial cell growth through regulation of autophagy in pulmonary hypertension [37]. *CEBPD* (cg16241714) encodes CCAAT/enhancer binding protein delta. *CEBPD* is involved in the regulation of apoptosis and cell proliferation and there is evidence that it might acts as tumour suppressor [38].

We compared the results of trans-ancestry analyses to previously reported EWAS of BP. Although the current study is smaller in size, we found evidence of some overlap between our results and the recent EWAS. Our study is the first blood pressure EWAS to our knowledge that has included South Asians, offering the chance to compare results from European and South Asian individuals.

There was no consistency in the magnitude and direction of associations comparing Europeans to South Asians, suggesting that peripheral blood DNA methylation patterns may differ between Europeans and South Asians in relation to blood pressure. This may reflect the fact that DNA methylation could index exposure to a different suite of risk factors in the two ethnic groups, or that different mechanisms contribute to the pathogenesis of

hypertension and its related phenotypic traits. Of note, the South Asian participants in the Southall and Brent REvisited (SABRE) study are first generation migrants, arriving in the UK as young adults. The potential early life and developmental antecedents of hypertension and blood pressure will therefore be considerably different between the two ethnic groups. This may explain to some extent the lack of consistency in methylation variable loci observed between the two ethnic groups. However, the study has limited statistical power due to the relatively modest sample size for these analyses and would benefit from replication in another sample of South Asian ancestry.

Where associations were observed, the effect sizes were modest in size between cases and controls at the identified CpG sites. Such differences are unlikely to have profound biological consequences but may in turn exert a polygenic-like effect, altering disease risk or trait characteristics by small amounts. Further work is required to understand the functional consequences of such subtle shifts in DNA methylation.

We carried out DMR analysis and identified a large number of DMRs for SBP and DBP in the trans-ancestry, European and South Asian subgroups. The analyses found support for some of the DMRs for CVD related traits in the literature including BP [17], myocardial infarction [18], dilated cardiomyopathy [19] and stroke [23]. There was some overlap between DMRs (mapped to genes) of SBP and DBP analyses conducted for South Asian and European groups. For SBP, there were three genes in common (*WRN*, *PTPRN2* and *CACNA1A*). There is evidence that *WRN* (Werner syndrome RecQ like helicase) protein plays an important role in DNA repair and in DNA replication[39, 40]. A previous study has shown that cells lacking *WRN* exhibit deletion of telomeres from single sister chromatids [41]. *PTPRN2* (protein tyrosine phosphatase, receptor type, N polypeptide 2) encodes a major autoantigen of relevance to type 1 diabetes [42, 43]. The *CACNA1A* (calcium voltage-gated channel subunit alpha1 A) is located on chromosome 19p13 that encodes the main subunit (1A) of the neuronal P/Q type voltage-gated calcium-ion channel[44]. Mutations in this gene have been associated with various neurological phenotypes [44]. For DBP, there was one DMR in common between South Asian and European groups (PRDM16). *PRDM16* (histone-lysine N-methyltransferase PRDM16) functions as a transcriptional regulator[45] and a previous study confirmed a causal role for this locus in human myocardial disease [19].

The strengths of this study include the study design where participants were from two different ethnicities; European and South Asian that were analysed collectively and individually using robust statistical methods. Additionally, the utilisation of HM450 arrays provided good coverage of the genome in terms of known annotated genes (although in total only covers <2% of all CpGs). The relatively modest sample size, the utilisation of only male participants and measurement of BP twice are among the limitations of this work.

## Conclusion

In conclusion, we identified associations between methylation and DBP across trans-ancestry and European-specific analyses. Lack of associations identified in South Asian specific analyses indicates that the associations between methylation and blood pressure may be different between European and South Asian populations.

## Methods

### Participant's information

SABRE is a population-based cohort including 4,857 people of European, South Asian and African Caribbean origin aged 40 to 69 living in West London, UK [46]. Peripheral blood samples were collected from the Southall participants at baseline (1988–91) for DNA extraction.

**Table 3. Distributions of study characteristics included in the EWAS analysis of SBP, DBP and hypertension.**

|  | Trans-ancestry | European | South Asian | P-value |
|---|---|---|---|---|
| **N** | 712(100%) | 364(51.12%) | 348(48.88%) |  |
| **Hypertension cases** | 126(100%) | 54(42.86%) | 72(57.14%) | 0.02* |
| **SBP** | 122.7(10.6) | 122.0(10.7) | 123.5(10.3) | 0.24 |
| **DBP** | 78.4(17.4) | 76.8(18.1) | 80.0(16.6) | $8.3 \times 10^{-5}$ |
| **Age (years) Mean (SD)** | 51(7.1) | 52(7.2) | 51(7.0) | 0.06 |
| **BMI (kg/m²) mean (SD)** | 25.8(3.5) | 26.0(3.3) | 25.7(3.6) | 0.3 |
| **Smokers (ever smokers)** | 361(100%) | 112(31.02%) | 266(73.68%) | <0.001* |
| **Social class (Manual)** | 473(100%) | 264(55.81%) | 231(48.84%) | 0.15* |

P-value: t-test for differences between European and South Asian groups and

*P-value = Fisher's exact test.

In the current analysis, 800 (400 European and 400 South Asian) samples from the SABRE cohort were randomly selected from available baseline samples of good DNA quality. Some samples were removed from the data set during quality control procedures and some samples were excluded due to missing information. After exclusion, 712 (364 European and 348 South Asian) individuals remained. We utilised EPISTRUCTURE[47] to conduct principal component analysis on a subset of 4913 genetically informative methylation probes. We were able to clearly differentiate between self-reported European and South Asian individuals. We did not identify any evidence of population substructure or admixture within either self-reported ancestral group.

These 712 individuals did not have known diabetes or coronary heart disease at baseline and were stratified by four-year age group and ethnicity. The SABRE study predominantly focused on the recruitment of men[46], for that reason epigenetic analyses were restricted to male participants.

Ethnicity in the SABRE cohort was assigned based on grand-parental origins from participant questionnaire. Blood pressure was measured on one occasion (the average of 2 consecutive readings) in the baseline research clinic as described previously [46].

All participants gave written informed consent. Approval for the baseline study was obtained from Ealing, Hounslow and Spelthorne, Parkside and University College London research ethnics committees. Characteristics of participants are shown in Table 3.

## Traits of interest

We investigated SBP, DBP and hypertension as our traits of interest. Hypertension was defined as occurrence of SBP≥140 mm Hg and/or DBP≥90 mm Hg, or receiving medication for hypertension as described previously [46]. The BP protocol was based on the INTERSALT study protocol [48].

## Covariates

Models were adjusted for age (years), body mass index (BMI) (kg/m²), ethnicity (European or South Asian for trans-ancestry analyses), smoking status (never or ever smoking) and social class (manual or non-manual occupation). Age, ethnicity, smoking status and social class of participants were collected from questionnaires. BMI was calculated from clinic measures of height and weight.

## DNA methylation and pre-processing

DNA was bisulfite converted using the Zymo EZ DNA Methylation™ kit (Zymo, Irvine, CA). Following conversion, DNA methylation was measured using the HM450 BeadChip in line with standard protocols at the University of Bristol, UK [49].

Samples failing QC (average probe detection p value≥0.01) were repeated and if unsuccessful excluded from further analysis. BeadChip intensity data were converted to β-values using the minfi package [50] in the R statistical programming language. Methylation beta values range from 0 (no cytosine methylation) to 1 (completely cytosine methylated). Raw beta values were normalised using the Functional Normalisation method of the minfi package [51]. We excluded control probes (n = 65), any probes with a detection p-value > 0.05 in more than 5% samples, non-CpG probes, polymorphic probes (defined as SNP-overlapping probes, probes with a SNP at the target CpG site, or probes with a SNP at the base next to the target CpG) and probes with a minor-allele frequency (MAF) ≥ 5%; based on UCSC common SNPs track for dbSNP build 137. We further excluded probes that are considered as cross-hybridizing [52]. We applied this stringent CpG filtering because polymorphic and cross-hybridizing probes can interfere with accurate detection of methylation levels. After excluding these features, 402,331 probes remained for the analysis.

## Estimation of cell counts

Cell count estimates were derived using the reference-based Houseman method [53] in the R minfi package [50] using the Reinius *et al*. dataset as reference [54]. This method estimates the relative proportions of six white blood cell subtypes (CD4+ T-lymphocytes, CD8+ T-lymphocytes, NK (natural killer) cells, B-lymphocytes, monocytes and granulocytes).

## Statistical analyses

**Epigenome-wide association study (EWAS) analysis.** We considered a trans-ancestry analysis as the primary model because this provided maximal statistical power by enabling us to include all participants. Our primary hypothesis was that the potential epigenetic mechanisms of elevated blood pressure would be different across these two ethnicities. However, we postulated that these differences may be common to key loci, rather than being at completely different loci.

Three sets of EWAS analyses were run to identify CpG sites associated with either SBP, DBP or hypertension. Normalised, untransformed beta-values, which are on a scale of 0 (completely unmethylated) to 1 (completely methylated) were utilised. Multiple linear regression models were run to evaluate the association between traits of interest and DNA methylation. Analyses were run for each ethnic subgroup and with all samples combined in a trans-ancestry analysis. To remove potential batch effects, principal components (PCs) were generated from methylation beta values using Principal Component Analysis (PCA) [55–57]. The PCs were generated for trans-ancestry and for individual ethnicities separately. The first four PCs were included as covariates in each of the EWAS. These PCs were not associated with exposure (i.e. SBP, DBP and hypertension) and captured 27%,27% and 28% of the variance for trans-ancestry, Europeans and South Asians respectively.

Two models were run for each trait of interest: i) an unadjusted model and ii) a model adjusted for confounders (age, BMI, ethnicity (where appropriate), smoking status, social class (occupational status according to the 1980 Registrar Generals classification of occupation; non manual or manual)), estimated cell counts (n = 6) and PCs. We adjusted for the confounders which we *a priori* selected on the basis that it was plausible they would influence blood pressure and blood DNA methylation[58].

The genomic inflation factor (lambda) was computed and Manhattan plots were generated to compare the genome wide distribution of p-values in EWAS. CpG sites with Bonferroni-corrected p-value<$1.24\times10^{-07}$ were considered to be associated with the trait of interest. We considered the fully adjusted trans-ancestry models as the primary analysis models.

**Differentially methylated region analysis.** In addition to EWAS analyses, differentially methylated region (DMR) analyses in relation to SBP, DBP and hypertension were conducted separately using the R package DMRcate [59]. In the DMR analysis, normalised, untrans-formed beta-values were used and the models were adjusted for confounders, estimated cell counts and PCs. DMRcate groups associated probes into separate DMRs if the gap between nucleotides is ≥1000 base pairs. P-values for associations were adjusted for multiple testing using the BH method and by the DMRcate algorithm.

Pathway analyses were carried out using the reactome (https://reactome.org/) to test the genes mapped for DMRs for enrichment of certain biological pathways.

**Comparison with previous EWAS.** A large EWAS study of SBP and DBP was recently conducted in amongst individuals of European, African American and Hispanic/Latino ancestry (n = 17,101) [16]. The study conducted an overall meta-analysis of the discovery and replication cohorts and identified 126 CpG sites associated with BP after Bonferroni correction (p-value<$1.0 \times 10^{-7}$). We compared our EWAS results of SBP and DBP of trans-ancestry analysis to this previously published EWAS of SBP and DBP.

## Supporting information

**S1 File. This file has 25 supplementary tables.**
(XLSX)

**S1 Zip. File has results of EWAS of diastolic blood pressure for trans-ancestry analyses (i.e. both ethnic groups combined).**
(ZIP)

**S2 Zip. File has results of EWAS of systolic blood pressure for trans-ancestry analyses (i.e. both ethnic groups combined).**
(ZIP)

**S3 Zip. File has results of EWAS of hypertension for trans-ancestry analyses (i.e. both ethnic groups combined).**
(ZIP)

**S4 Zip. File has results of EWAS of diastolic blood pressure for South Asian analyses.**
(ZIP)

**S5 Zip. File has results of EWAS of systolic blood pressure for South Asian analyses.**
(ZIP)

**S6 Zip. File has results of EWAS of hypertension for South Asian analyses.**
(ZIP)

**S7 Zip. File has results of EWAS of diastolic blood pressure for European analyses.**
(ZIP)

**S8 Zip. File has results of EWAS of hypertension for European analyses.**
(ZIP)

**S9 Zip. File has results of EWAS of systolic blood pressure for European analyses.**
(ZIP)

## Author Contributions

**Conceptualization:** Nabila Kazmi, Hannah R. Elliott, Caroline L. Relton.

**Formal analysis:** Nabila Kazmi.

**Investigation:** Nabila Kazmi, Hannah R. Elliott.

**Methodology:** Nabila Kazmi, Hannah R. Elliott.

**Resources:** Therese Tillin, Alun D. Hughes, Nish Chaturvedi.

**Supervision:** Hannah R. Elliott, Tom R. Gaunt, Caroline L. Relton.

**Validation:** Nabila Kazmi.

**Writing – original draft:** Nabila Kazmi.

**Writing – review & editing:** Nabila Kazmi, Hannah R. Elliott, Kim Burrows, Therese Tillin, Alun D. Hughes, Nish Chaturvedi, Tom R. Gaunt, Caroline L. Relton.

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
