## [Decision Letter · Decision Letter 0]

4 Sep 2019

PONE-D-19-20995

Associations Between High Blood Pressure and DNA Methylation

PLOS ONE

Dear Dr Kazmi,

Thank you for submitting your manuscript to PLOS ONE. After careful consideration, we feel that it has merit but does not fully meet PLOS ONE’s publication criteria as it currently stands. Therefore, we invite you to submit a revised version of the manuscript that addresses the points raised during the review process.

We would appreciate receiving your revised manuscript by Oct 19 2019 11:59PM. To enhance the reproducibility of your results, we recommend that if applicable you deposit your laboratory protocols in protocols.io, where a protocol can be assigned its own identifier (DOI) such that it can be cited independently in the future. For instructions see: http://journals.plos.org/plosone/s/submission-guidelines#loc-laboratory-protocols

We look forward to receiving your revised manuscript.

Kind regards,

Heming Wang, PhD

Academic Editor

PLOS ONE

Journal Requirements:

"TRG receives funding from GlaxoSmithKline, Biogen and Sanofi for research unrelated to the work presented here."

Reviewers' comments:

Reviewer's Responses to Questions

**Comments to the Author**

1. Is the manuscript technically sound, and do the data support the conclusions?

Reviewer #1: Partly

Reviewer #2: Yes

2. Has the statistical analysis been performed appropriately and rigorously? 

Reviewer #1: Yes

Reviewer #2: Yes

3. Have the authors made all data underlying the findings in their manuscript fully available?

Reviewer #1: Yes

Reviewer #2: Yes

4. Is the manuscript presented in an intelligible fashion and written in standard English?

Reviewer #1: Yes

Reviewer #2: Yes

5. Review Comments to the Author

Reviewer #1: While this study analyzes an under represented group, the manuscript is a little unclear and could use a few more edits for readability. The authors identify one trans-specific site and a few European specific sites. They in addition found some DMR associated with SBP and DBP.

Comments

Major

----

-Did the authors examine if there was any genetic admixture within the South Asian participants that could impact the results? Perhaps look at some of the methods that have developed over past years for getting at population structure from DNA methylation.

-The authors mention the unadjusted results throughout the document but I do not think that this should be highlighted as for a result to pass EWAS threshold needs to be significant adjusting for all these known confounders.

-What does cg07598370 effect look like in each of the populations?

-Line 192: For the goodness of fit R2. There will likely be good concordance amongst the trans-ancestry and the respective ethnic specific analysis as those European participants or South-Asian participants were included in the trans-ancestry analysis. Would remove and just focus on the South-Asian to European Ancestry. Did the authors examine the spearman correlation? Or examine the correlation of the test statistics.

The document could use some grammatical work, just going through and checking the sentence structure throughout.

Minor

--

-Line 77: “We planned to analyze” Don’t say planned, say we analyzed. Keep tense consistent.

Line 83: Have brief paragraph describing the study participants and place Table 1 here. Otherwise Table 1 is referred to prior to Table 2.

Line 261- Reword sentence. Change “The” to “A”

Line 276-Get rid of second AFB

Line 281- Typo? “related protein likewise many other proteins”

Line 288- Do not say probably. Perhaps say “evidence associated with”.

Line 288- “CD59 (CD59, molecule (CD59, blood group), cg0635652)”. Reword/organize. Part is confusing.

Line 290- Reword “blood coagulation... among its biological pathways”. Reads as though has possession of those pathways. Perhaps say “Gene is known to be related to pathways of…”

Line 298- Reword. Say EWAS twice in one sentence.

Mention that first generation earlier in the document.

Line 348-350: Two repetitive sentences that the study is male.

Line 398: Statistical analysis. Is DNA methylation the dependent variable?

Reviewer #2: In this article, the authors examine the association of DNA methylation with hypertension and continuous blood pressure in both European-descent and South-Asian descent men. The sample size is fairly large, and the analytical plan is carefully described and conducted. The results provide data on a seldom described population in cardiovascular epigenetics that enriches this field of research.

My only major comment is to highlight and emphasis the results of the DMR analysis more. The results of the CpG site-specific analysis take center stage, especially when summarizing main conclusions in the abstract, discussion, and conclusion section and also in the main figures. The DMR analyses reveal quite a few long DMRs in interesting genes. Discovery of DMRs are potentially more informative than single CpG sites which alone may not be very biologically relevant. Can the authors dig into the DMRs more – perhaps providing a Venn Diagram of overlapping genes between the 3 analysis groups for DMRs? Or running a pathway analysis on the genes that show up as q<0.05 in the DMR analysis? Also please explain how q<0.05 is calculated for the DMRs in the methods (is this through the DMRcate algorithm or separately calculated?)

A few minor suggestions are also noted:

-Ln 86 to 91: When discussing these sites, it would be helpful to readers to put together the ones that are in the same genes. Same comment for Ln 137-144.

-Figures 1 and 2 should include indication of which CpG sites passed the significance cut-off used for this study (q<0.05) since the Bonferonni cut-off is mentioned used in the discussion or methods.

-Ln 263: Expand upon this (how do olfactory receptors regulate blood pressure by the kidney)?

-Overall proof-reading/editing of the discussion section is needed.

-Is there any longitudinal data on these participants or only cross-sectional blood pressure?

-Ln 415: What is considered the exposure here? Does that mean the outcome (e.g., blood pressure)?

-Ln 418-420: How were these covariates selected? What is the social class variable? Which cell types were included in the models (all 6 or a subset of them?)

-Table 3: Please denote if there is overlap between trans-ancestry and European and/or South Asian models in these genes lists (example: HLX. The authors could be a symbol by it that denotes overlap).

-Table 1 should include percentages along with numbers for the categorical variables.

6. PLOS authors have the option to publish the peer review history of their article (what does this mean?). If published, this will include your full peer review and any attached files.

Reviewer #1: No

Reviewer #2: Yes: Jaclyn Goodrich

---

## [Author Response · Author response to Decision Letter 0]

18 Oct 2019

Manuscript: Associations Between High Blood Pressure and DNA Methylation

Reviewer #1 comments and response from authors:

While this study analyzes an under represented group, the manuscript is a little unclear and could use a few more edits for readability. The authors identify one trans-specific site and a few European specific sites. They in addition found some DMR associated with SBP and DBP.

Specific comments: 

Comment 1: Did the authors examine if there was any genetic admixture within the South Asian participants that could impact the results? Perhaps look at some of the methods that have developed over past years for getting at population structure from DNA methylation.

Response: We thank the reviewer for this comment. We have utilised EPISTRUCTURE (https://epigeneticsandchromatin.biomedcentral.com/articles/10.1186/s13072-016-0108-y) to conduct principal component analysis on a subset of 4913 genetically informative methylation probes. We were able to clearly differentiate between self-reported European and South Asian individuals. We did not identify any evidence of population substructure or admixture within either self-reported ancestral group. We have added few sentences to the main text of the paper to communicate this point.

“We utilised EPISTRUCTURE(54) to conduct principal component analysis on a subset of 4913 genetically informative methylation probes. We were able to clearly differentiate between self-reported European and South Asian individuals. We did not identify any evidence of population substructure or admixture within either self-reported ancestral group.” (page:19; lines: 401-405)

Comment 2: The authors mention the unadjusted results throughout the document but I do not think that this should be highlighted as for a result to pass EWAS threshold needs to be significant adjusting for all these known confounders.

Response: We agree with reviewer and have deleted the unnecessary text emphasising unadjusted EWAS results. Please see the Results section; pages 5-8.

Comment 3: What does cg07598370 effect look like in each of the populations?

Response: The mean (SD) of this CpG in European is 0.84 (0.07) and in South Asian population is 0.83 (0.05). We have added this information to the manuscript:

 “However, one additional CpG site (cg07598370 near OR5AP2) was identified in the adjusted EWAS, below the Bonferroni-corrected threshold (Figure 1 and Supplementary File 1, Table S4). DBP was associated with lower DNA methylation at this CpG site with a 0.1% decrease in DNA methylation per 1 mmHg increase in DBP. The mean and standard deviation (SD) of DNA methylation at this CpG site in the European group was 0.84 (0.07) and in the South Asian group was 0.83 (0.05).” (page:6; lines:111-117)

Comment 4: Line 192: For the goodness of fit R2. There will likely be good concordance amongst the trans-ancestry and the respective ethnic specific analysis as those European participants or South-Asian participants were included in the trans-ancestry analysis. Would remove and just focus on the South-Asian to European Ancestry. Did the authors examine the spearman correlation? Or examine the correlation of the test statistics.

Response: We recognise this duplication in data sources when considering this issue and have now removed the comparisons made between trans-ancestry and the European and South Asian ancestry groups but kept the comparisons made between individual ethnicities. 

In answer to second point, there was a linear fit between European and South Asian participants data. We have now revised the text in the manuscript accordingly:

“The magnitude of associations between European and South Asian populations taken from the fully adjusted EWAS of SBP, DBP and hypertension were compared to each other using a linear fit and a random intercept and slope multilevel model in a pair-wise comparison.” (page 10; lines: 205-208)

Minor comments

The document could use some grammatical work, just going through and checking the sentence structure throughout.

Response: We have revised the grammar throughout the manuscript.

Comment 1: Line 77: “We planned to analyze” Don’t say planned, say we analyzed. Keep tense consistent.

Response: We have revised the sentence as suggested:

 “We analyzed all samples together in a trans-ancestry analysis, then conducted analyses in each ethnic group separately.” (page 4; lines: 81-82)

Comment 2: Line 83: Have brief paragraph describing the study participants and place Table 1 here. Otherwise Table 1 is referred to prior to Table 2.

Response: Thank you for pointing out this mistake. We have now corrected the order of the Tables and their referral. 

Comment 3: Line 261- Reword sentence. Change “The” to “A”

Response: We have revised the sentence accordingly:

“A genetic variant near the olfactory receptor, family 5, subfamily AP, member 2 (OR5AP2) has been previously reported to be associated with hematological phenotypes (27), in addition olfactory receptors are known to regulate blood pressure via their renal expression (28).” (page:14; lines: 293-97)

Comment 4: Line 276-Get rid of second AFB

Response: We believe that the reviewer meant AFAP, we have removed the second occurrence of this from the revised draft:

“One CpG site (cg04751533), in the gene body of AFAP (actin filament-associated protein), acts as an actin-binding and crosslinking protein and is enriched in SRC and phorbol ester induced podosomes (35).” (page: 15; lines: 311-314)

Comment 5: Line 281- Typo? “related protein likewise many other proteins”

Response: We have reworded the sentence:

 “ATG101 is an autophagy related protein, with relevance to blood pressure because autophagy plays a key role in pulmonary vascular remodelling via regulation of apoptosis and hyperproliferation of pulmonary arterial endothelial cells (37).”(page: 15; lines: 317-321)

Comment 6: Line 288- Do not say probably. Perhaps say “evidence associated with”.

Response: The sentence has been revised:

 “CEBPD is involved in the regulation of apoptosis and cell proliferation and there is evidence that it might acts as tumour suppressor (38).” (page: 15; lines: 323-325)

Comment 7: Line 288- “CD59 (CD59, molecule (CD59, blood group), cg0635652)”. Reword/organize. Part is confusing.

Response: This sentence has been removed from the revised draft.

Comment 8: Line 290- Reword “blood coagulation... among its biological pathways”. Reads as though has possession of those pathways. Perhaps say “Gene is known to be related to pathways of…”

Response: This sentence has been removed from the revised draft.

Comment 9: Line 298- Reword. Say EWAS twice in one sentence.

Response: The sentence has been revised: 

“We compared the results of trans-ancestry analyses to previously reported EWAS of BP.” (page: 16; line:334-335)

Comment 10: Mention that first generation earlier in the document.

Response: We believe the reviewer is referring to the section: “Of note, the South Asian participants in the Southall and Brent REvisited (SABRE) study are first generation migrants, arriving in the UK as young adults. The potential early life and developmental antecedents of hypertension and blood pressure will therefore be considerably different between the two ethnic groups.”

This is the first time that SABRE was mentioned in the document as, per PLOS guidelines, the Methods section is placed after the Discussion. We have inserted a short sentence in the Abstract to alert readers to this fact much earlier in the manuscript:

 “We analyzed samples from 364 European and 348 South Asian men (first generation migrants to the UK) from the Southall And Brent REvisited cohort..” (page:2; lines: 43-44)

Comment 11: Line 348-350: Two repetitive sentences that the study is male.

Response: We have removed the sentence “All individuals were male” to avoid repetition. (page 19)

Comment 12: Line 398: Statistical analysis. Is DNA methylation the dependent variable?

Response: Yes, DNA methylation is a dependent variable.

Reviewer #2 comments and response from authors:

In this article, the authors examine the association of DNA methylation with hypertension and continuous blood pressure in both European-descent and South-Asian descent men. The sample size is fairly large, and the analytical plan is carefully described and conducted. The results provide data on a seldom described population in cardiovascular epigenetics that enriches this field of research.

Specific comments: 

My only major comment is to highlight and emphasis the results of the DMR analysis more. The results of the CpG site-specific analysis take center stage, especially when summarizing main conclusions in the abstract, discussion, and conclusion section and also in the main figures. The DMR analyses reveal quite a few long DMRs in interesting genes. Discovery of DMRs are potentially more informative than single CpG sites which alone may not be very biologically relevant. Can the authors dig into the DMRs more – perhaps providing a Venn Diagram of overlapping genes between the 3 analysis groups for DMRs? Or running a pathway analysis on the genes that show up as q<0.05 in the DMR analysis? Also please explain how q<0.05 is calculated for the DMRs in the methods (is this through the DMRcate algorithm or separately calculated?)

Response: We thank the reviewer for this comment. We have now provided more details about the DMR analyses. We have created Venn diagrams to show the overlap between the DMRs identified in the different ethnic groups and have revised the manuscript accordingly.

Results:

“We created Venn diagrams to identify overlap between DMRs (mapped to genes) of SBP and DBP analyses conducted for South Asian and European groups (Figure 4). For SBP, there were three genes in common (WRN, PTPRN2 and CACNA1A). For DBP, there was one DMR in common between South Asian and European groups (PRDM16).

Figure 4. Venn diagrams to identify overlap between DMRs (mapped to genes) of DBP (A) and SBP (B) analyses conducted for South Asian and European groups. 

Figure 4. SA denotes South Asian and Eu denotes European ancestry.” (pages: 12-13; lines:251-258)

Discussion:

“There was some overlap between DMRs (mapped to genes) of SBP and DBP analyses conducted for South Asian and European groups. For SBP, there were three genes in common (WRN, PTPRN2 and CACNA1A). There is evidence that WRN (Werner syndrome RecQ like helicase) protein plays an important role in DNA repair and in DNA replication(46, 47). A previous study has shown that cells lacking WRN exhibit deletion of telomeres from single sister chromatids(48). PTPRN2 (protein tyrosine phosphatase, receptor type, N polypeptide 2) encodes a major autoantigen of relevance to type 1 diabetes (49, 50). The CACNA1A (calcium voltage-gated channel subunit alpha1 A) is located on chromosome 19p13 that encodes the main subunit (1A) of the neuronal P/Q type voltage‐gated calcium‐ion channel(51). Mutations in this gene have been associated with various neurological phenotypes (51). For DBP, there was one DMR in common between South Asian and European groups (PRDM16). PRDM16 (histone-lysine N-methyltransferase PRDM16) functions as a transcriptional regulator(52) and a previous study confirmed a causal role for this locus in human myocardial disease (19). ” (pages: 17-18; lines: 361-376)

In answer to “please explain how q<0.05 is calculated for the DMRs in the methods (is this through the DMRcate algorithm or separately calculated?)”

Response: Yes, it was calculated within the DMRcate algorithm. We have added a sentence to the Methods section: 

 “P-values for associations were adjusted for multiple testing using the BH method and by the DMRcate algorithm.” (page 23; lines:497-498)

Minor comments

Comment 1: Ln 86 to 91: When discussing these sites, it would be helpful to readers to put together the ones that are in the same genes. Same comment for Ln 137-144.

Response: In response to Reviewer 1’s comment (comment #2) we have deleted these lines from the revised draft.

Comment 2: Figures 1 and 2 should include indication of which CpG sites passed the significance cut-off used for this study (q<0.05) since the Bonferonni cut-off is mentioned used in the discussion or methods.

Response: We have revised the legends of the figures indicating that CpGs that were above the red line were considered to be associated with the trait. We have revised the text in Results section accordingly.

 “The blue line is drawn to separate the CpG sites that surpassed p-value<1×10-05, a threshold for a suggestive association and the red line to separate the CpG sites that surpassed the Bonferroni-corrected threshold (p-value<1.24×10-07). The CpG sites that surpassed the Bonferroni-corrected threshold were considered to be associated with the trait.” 

Comment 3: Ln 263: Expand upon this (how do olfactory receptors regulate blood pressure by the kidney)?

Response: We have provided more detail:

“A genetic variant near the olfactory receptor, family 5, subfamily AP, member 2 (OR5AP2) has been previously reported to be associated with hematological phenotypes (27), in addition olfactory receptors are known to regulate blood pressure via their renal expression (28). Evidence has shown that renal sensory receptors play an important role in blood pressure regulation and olfactory receptors belonging to this group of sensory receptors(29).” (page: 14; lines: 293-99)

Comment 4: Overall proof-reading/editing of the discussion section is needed.

Response: We have proof-read and corrected the grammar throughout the manuscript.

Comment 5: Is there any longitudinal data on these participants or only cross-sectional blood pressure?

Response: The SABRE cohort underwent a follow up clinic approximately 17 years after the baseline study. We collected methylation profiles on <20% of follow up samples corresponding with the baseline samples included in our main analysis. Because the sample size was small (therefore limiting statistical power), we did not conduct longitudinal analyses on these data.

Comment 6: Ln 415: What is considered the exposure here? Does that mean the outcome (e.g., blood pressure)?

Response: We apologise for any confusion; our outcome is DNA methylation and exposures are systolic, diastolic blood pressure and hypertension. We have revised the text in the manuscript to make this clearer:

“The first four PCs were included as covariates in each of the EWAS. These PCs were not associated with exposure (i.e. SBP, DBP and hypertension) and captured 27%,27% and 28% of the variance for trans-ancestry, Europeans and South Asians respectively.” (pages: 22-23; lines: 475-478)

Comment 7: Ln 418-420: How were these covariates selected? What is the social class variable? Which cell types were included in the models (all 6 or a subset of them?)

Response: We adjusted for the confounders which we a priori selected on the basis that it was plausible they would influence blood pressure and blood DNA methylation.

Social class represents occupational status according to the 1980 Registrar General’s classification of occupation (i.e. non manual or manual) and all six cell types were included in the model.

We have revised the text in the manuscript accordingly:

 “Two models were run for each trait of interest: i) an unadjusted model and ii) a model adjusted for confounders (age, BMI, ethnicity (where appropriate), smoking status, social class (occupational status according to the 1980 Registrar Generals classification of occupation; non manual or manual)), estimated cell counts (n=6) and PCs. We adjusted for the confounders which we a priori selected on the basis that it was plausible they would influence blood pressure and blood DNA methylation(65).” (page: 23; lines: 460-65)

Comment 8 -Table 3: Please denote if there is overlap between trans-ancestry and European and/or South Asian models in these genes lists (example: HLX. The authors could be a symbol by it that denotes overlap).

Response: We have denoted the overlap between results of trans-ancestry and European analyses by * and between trans-ancestry and South Asian by **. We have provided this information in the legend of Table 2:

“* when a gene is common between trans-ancestry and European analyses, ** when a gene is common between trans-ancestry and South Asian analyses.” (page: 12; lines: 241-242)

Comment 9 -Table 1 should include percentages along with numbers for the categorical variables.

Response: We have provided the percentages. In the revised manuscript Table 1 is now Table 3. (page: 20)

---

## [Decision Letter · Decision Letter 1]

12 Nov 2019

PONE-D-19-20995R1

Associations Between High Blood Pressure and DNA Methylation

PLOS ONE

Dear Dr Kazmi,

Thank you for submitting your manuscript to PLOS ONE. After careful consideration, we feel that it has merit but does not fully meet PLOS ONE’s publication criteria as it currently stands. Therefore, we invite you to submit a revised version of the manuscript that addresses the points raised during the review process.

We would appreciate receiving your revised manuscript by Dec 27 2019 11:59PM. To enhance the reproducibility of your results, we recommend that if applicable you deposit your laboratory protocols in protocols.io, where a protocol can be assigned its own identifier (DOI) such that it can be cited independently in the future. For instructions see: http://journals.plos.org/plosone/s/submission-guidelines#loc-laboratory-protocols

We look forward to receiving your revised manuscript.

Kind regards,

Heming Wang, PhD

Academic Editor

PLOS ONE

Reviewers' comments:

Reviewer's Responses to Questions

**Comments to the Author**

1. If the authors have adequately addressed your comments raised in a previous round of review and you feel that this manuscript is now acceptable for publication, you may indicate that here to bypass the “Comments to the Author” section, enter your conflict of interest statement in the “Confidential to Editor” section, and submit your "Accept" recommendation.

Reviewer #1: (No Response)

Reviewer #2: All comments have been addressed

2. Is the manuscript technically sound, and do the data support the conclusions?

Reviewer #1: Yes

Reviewer #2: Yes

3. Has the statistical analysis been performed appropriately and rigorously? 

Reviewer #1: Yes

Reviewer #2: Yes

4. Have the authors made all data underlying the findings in their manuscript fully available?

Reviewer #1: Yes

Reviewer #2: Yes

5. Is the manuscript presented in an intelligible fashion and written in standard English?

Reviewer #1: Yes

Reviewer #2: Yes

6. Review Comments to the Author

Reviewer #1: The authors have sufficiently addressed my previous questions and the paper is much improved. I only have a few remaining minor comments/questions.

--Line 187. Differentially methylated region analysis.

I know the authors found four regions of overlap, but I was wondering if the authors looked to see if the other discrepant regions happened to fall in similar pathways?

--Line 216.

This is more of a formatting issue. "Known genetic variants in DMRs" is on the same line as Figure 4 legend.

--Line 370

When were the traits measured? Were they also at baseline?

Reviewer #2: (No Response)

7. PLOS authors have the option to publish the peer review history of their article (what does this mean?). If published, this will include your full peer review and any attached files.

Reviewer #1: No

Reviewer #2: Yes: Jaclyn Goodrich

---

## [Author Response · Author response to Decision Letter 1]

18 Dec 2019

Manuscript: Associations Between High Blood Pressure and DNA Methylation

Reviewer #1: 

Comment 1: Line 187. Differentially methylated region analysis. I know the authors found four regions of overlap, but I was wondering if the authors looked to see if the other discrepant regions happened to fall in similar pathways?

Response: Now, we have performed the pathway analyses and revised the manuscript accordingly. 

Results:

“We performed pathway analyses for the genes mapped for DMRs of SBP and DBP analyses conducted for South Asian and European groups. For SBP analyses conducted for Europeans and DBP analyses conducted for South Asians, we identified pathways enriched with FDR q-value<0.05. The identified pathways for SBP analyses conducted for Europeans were NOTCH2 intracellular domain regulates transcription and NOTCH4 intracellular domain regulates transcription. The identified pathway for DBP analyses conducted for South Asians was insulin-like growth factor-2 mRNA binding proteins (IGF2BPs/IMPs/VICKZs) bind RNA.”(pages:11-12; lines: 217-224)

Methods:

“Pathway analyses were carried out using the reactome (https://reactome.org/) to test the genes mapped for DMRs for enrichment of certain biological pathways.”(page:22; lines: 452-453)

Comment 2: Line 216. This is more of a formatting issue. "Known genetic variants in DMRs" is on the same line as Figure 4 legend.

Response: We have corrected the formatting issue. Please see line 226 on page 12.

Comment 3: Line 370. When were the traits measured? Were they also at baseline?

Response: Yes, please see:

“Blood pressure was measured on one occasion (the average of 2 consecutive readings) in the baseline research clinic as described previously (46).” (pages 17-18; lines: 366-367)

---

## [Decision Letter · Decision Letter 2]

30 Dec 2019

Associations Between High Blood Pressure and DNA Methylation

PONE-D-19-20995R2

Dear Dr. Kazmi,

We are pleased to inform you that your manuscript has been judged scientifically suitable for publication and will be formally accepted for publication once it complies with all outstanding technical requirements.

With kind regards,

Heming Wang, PhD

Academic Editor

PLOS ONE

Additional Editor Comments (optional):

Reviewers' comments:

Reviewer's Responses to Questions

**Comments to the Author**

1. If the authors have adequately addressed your comments raised in a previous round of review and you feel that this manuscript is now acceptable for publication, you may indicate that here to bypass the “Comments to the Author” section, enter your conflict of interest statement in the “Confidential to Editor” section, and submit your "Accept" recommendation.

Reviewer #1: (No Response)

2. Is the manuscript technically sound, and do the data support the conclusions?

Reviewer #1: Yes

3. Has the statistical analysis been performed appropriately and rigorously? 

Reviewer #1: Yes

4. Have the authors made all data underlying the findings in their manuscript fully available?

Reviewer #1: (No Response)

5. Is the manuscript presented in an intelligible fashion and written in standard English?

Reviewer #1: Yes

6. Review Comments to the Author

Reviewer #1: The authors have addressed all of my comments. Minor formatting/grammar comment: Line 145: "theEWAS"-> "the EWAS".

7. PLOS authors have the option to publish the peer review history of their article (what does this mean?). If published, this will include your full peer review and any attached files.

Reviewer #1: No

---

## [Editor Report · Acceptance letter]

13 Jan 2020

PONE-D-19-20995R2 

Associations Between High Blood Pressure and DNA Methylation 

Dear Dr. Kazmi:

I am pleased to inform you that your manuscript has been deemed suitable for publication in PLOS ONE. Congratulations! Your manuscript is now with our production department. 

With kind regards,

on behalf of

Dr. Heming Wang 

Academic Editor

PLOS ONE